# Hunt For The Unique, Stable, Sparse And Fast Feature Learning On Graphs

**Saurabh Verma**
Department of Computer Science
University of Minnesota, Twin Cities
`verma@cs.umn.edu`

**Zhi-Li Zhang**
Department of Computer Science
University of Minnesota, Twin Cities
`zhang@cs.umn.edu`

## Abstract

For the purpose of learning on graphs, we hunt for a graph feature representation that exhibit certain uniqueness, stability and sparsity properties while also being amenable to fast computation. This leads to the discovery of *family of graph spectral distances* (denoted as FGSD) and their based graph feature representations, which we prove to possess most of these desired properties. To both evaluate the quality of graph features produced by FGSD and demonstrate their utility, we apply them to the graph classification problem. Through extensive experiments, we show that a simple SVM based classification algorithm, driven with our powerful FGSD based graph features, significantly outperforms all the more sophisticated state-of-art algorithms on the unlabeled node datasets in terms of both accuracy and speed; it also yields very competitive results on the labeled datasets – despite the fact it does *not* utilize any node label information.

## 1   Introduction

In the past decade, there has been tremendous interests in learning on collection of graphs for various purposes, in particular for solving graph classification problem. Several applications of graph classification can be found in the domain of bioinformatics, or chemoinformatics, or social networks. A fundamental question inherent in graph classification is determining whether two graph structures are identical, i.e., the graph isomorphism problem, which was not known to belong either P or NP until recently. In the seminal paper [2], Babai shows that the graph isomorphism can be solved in quasipolynomial time; while of enormous theoretical signficance, the implication of this result in developing practical algorithms is still unclear. Fortunately, in graph classification problems one is more interested in whether two graphs have "similar" (as opposed to *identical*) structures. This allows for potentially much faster (yet not fully explored) algorithms to be successfully applied to the graph classification while also accounting for graph isomorphism. One approach to get around both these *intimately tied problems together* is to learn an explicit graph representation that is invariant under graph isomorphism[1] but also useful for extracting graph features.

More specifically, given a graph $G$, we are interested in learning a graph representation (or spectrum), $\mathcal{R} : G \rightarrow (g_1, g_2, ..., g_r)$, that captures certain inherent "atomic" (unique) sub-structures of the graph and is invariant under graph isomorphism (i.e., two isomorphic graphs yield the same representation). Subsequently, we want to learn a feature function $\mathcal{F} : \mathcal{R} \rightarrow (f_1, f_2, ..., f_d)$ from $\mathcal{R}$ such that the graph features $\{f_i\}_{i=1}^d$ can be employed for solving the graph classification problem. However, in machine learning, not much attention has been given towards learning $\mathcal{R}$ and most of the previous studies have focused on designing graph kernels and thus bypasses computing any explicit graph representation. The series of papers (19, 20, 22) by Kondor et al. are some of the first (and few) that are concerned with constructing explicit graph features – using a group theoretic approach – that are invariant to graph isomorphism and can be successfully applied to the graph classification problem.

Figure 1: Graph Generation Model: Graph spectrum is assumed to be encoded in pairwise node distances which are generated from some distribution. Nodes connect together to form a graph in such a way that pairwise node distances are preserved (eg. (●—●) node-pair with distance 0.75 is preserved even though they are not directly connected).

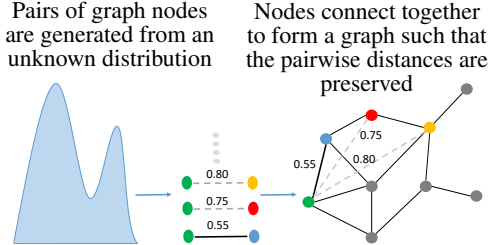

Pairs of graph nodes are generated from an unknown distribution

Nodes connect together to form a graph such that the pairwise distances are preserved

Inspired by such an approach, we also explicitly deal with learning a graph representation $\mathcal{R}$ and show how to derive graph features $\mathcal{F}$ from $\mathcal{R}$.

Our approach is quite novel and builds upon the following assumption: *Graph atomic structure (or spectrum) is encoded in the multiset[2] of all node pairwise distances.* Figure 1 shows the complete graph generation model based on this premise. The origin of our assumption can be traced back to the study of homometric structure, i.e, structures with the same multiset of interatomic distances [28]. On graphs, two vertex sets are called non-homometric if the multisets of distances determined by them are different. (It is an unexplored problem whether there exists any distance metric on the graph for which two vertex sets of non-isomorphic graphs are always non-homometric; but the converse is not true, an example is the shortest path distance.) This argument provides the *validity* of our assumption that the graph atomic structure is being encoded in pairwise distances. Further, we have empirically found that the biharmonic distance [23] multisets are unique for at-least upto 10-vertex size simple connected graphs ($\sim$ 11 million graphs) and it remains as an *open problem* to show a contradictory example. Moreover, we show that for a certain distance function $\mathcal{S}_f$ on the graph, one can *uniquely recover* all the graph intrinsic properties while also being able to capture both local & global information about the graph. Thus, we define $\mathcal{R}$ as the multiset of node pairwise distances based on some distance function $\mathcal{S}_f$, which will be the main focus of this paper.

We hunt for such a family of distances on graphs and its core members for which most of the properties of an ideal graph spectrum (see Section 3) hold, including invariance under graph isomorphism and the uniqueness property. This hunt leads us to the discovery of a family of graph spectral distance (FGSD) and one would find harmonic (effective resistance) and biharmonic distance on graphs as the suitable members of this family for graph representation $\mathcal{R}$. Finally, for solving graph classification (where graphs can be of different nodes sizes), we simply construct $\mathcal{F}$ feature vector from the histogram of $\mathcal{R}$ (a multiset) and feed it to a standard classification algorithm.

Our current work focuses only on unlabeled graphs but can be extended to labeled graphs using the same strategy as in shortest path kernel [4]. Nevertheless, our comprehensive results show that FGSD graph features are powerful enough to significantly outperform the current state-of-art algorithms on unlabeled datasets and are very competitive on labeled datasets – despite the fact that they do *not* utilize any node label information. **In summary, the major contributions of our paper are:**

- Introducing a *novel & conceptually simple yet powerful* graph feature representation (or spectrum) based on the multiset of node pairwise distances.
- Discovering FGSD as a well-suited candidate for our proposed graph spectrum.
- Proving that FGSD based graph features exhibit certain uniqueness, stability, sparsity properties and can be computationally fast with $\mathcal{O}(N^2)$ complexity, where $N$ is the number of graph nodes in a graph.
- Showing the superior performance of FGSD based graph features on graph classification tasks.

## 2    Related Work

Previous studies on graph classification can be grouped into three main categories. The first category is concerned with constructing explicit graph features such as the skew spectrum 20 and its successor, graphlet spectrum [22] based on group-theoretic approaches. Both are computational expensive. The second and more popular category deals with designing graph kernels, among which, strong ones are graphlets [30], random walks or shortest paths [4], neighborhood subgraph pairwise distance

kernel [9], Weisfeiler-Lehman kernel [31], deep graph kernels [34], graph invariant kernels [27] and multiscale Laplacian graph kernel [21]. A tangential work [24] related to constructing features based on atoms 3D space coordinates rather than operating on a graph structure, can be also considered in this category. Our effort on learning $\mathcal{R}$ from FGSD can be seen as a part of first category, since we explicitly investigate numerous properties of our proposed graph spectrum. While, extracting $\mathcal{F}$ from $\mathcal{R}$ is more inspired from the work of graph kernels.

The third category involves developing convolutional neural networks (CNNs) for graphs, where several models have been proposed to define convolution networks on graphs. The most common model is based on generalizing convolutional networks through the graph Fourier transform via a graph Laplacian [7, 16]. Defferrard et al. [11] extend this model by constructing fast localized spectral filters for efficient graph coarsening as a pooling operation for CNNs on graphs. Some variants of these models were considered in [18, 1], where the output of each neural network layer is computed using a propagation rule that takes the graph adjacency matrix and node feature vectors into account while updating the network weights. In [12], the convolution operation is defined by hashing of local graph node features along with the local structure information. Likewise, in [26] local node sequences are "canonicalized" to create receptive fields and then fed into a 1D convolutional neural network for classification. Among the aforementioned graph CNNs models, **only those in** [26, 1, 12] are relevant to this work since they are designed to account for graphs of different sizes, while others assume a global structure where the one-to-one correspondence of input vertices are already known.

## 3   Family of Graph Spectral Distances and Graph Spectrum

**Basic Setup and Notations**: Consider a weighted, undirected (and *connected*) graph $G = (V, E, W)$ of size $N = |V|$, where $V$ is the vertex set, $E$ the edge set (with no self-loops) and $W = [w_{xy}]$ the nonnegative *weighted* adjacency matrix. The standard graph Laplacian is defined as $L = D - W$, where $D$ is the degree matrix. It is semi-definite and admits an eigen-decomposition of the form $L = \mathbf{\Phi}\Lambda\mathbf{\Phi}^T$, where $\Lambda = \mathbf{diag}[\lambda_k]$ is the diagonal matrix formed by the eigenvalues $\lambda_0 = 0 < \lambda_1 \leq \cdots \leq \lambda_{N-1}$, and $\mathbf{\Phi} = [\phi_0, ..., \phi_{N-1}]$ is an orthogonal matrix formed by the corresponding eigenvectors $\phi_k$'s. For $x \in V$, we use $\phi_k(x)$ to denote the $x$-entry value of $\phi_k$. Let $f$ be an arbitrary nonnegative (real-analytical) function on $R^+$ with $f(0) = 0$, $\mathbf{1} = [1, .., 1]^T$ is the all-one vector and $J = \mathbf{1}\mathbf{1}^T$. Then, using slight abuse of notion, we define $f(L) := \mathbf{\Phi} f(\Lambda)\mathbf{\Phi^T}$ and $f(\Lambda) := \mathbf{diag}[f(\lambda_k)]$. Also, $f(L)_{xy}$ represent $xy$-entry value in $f(L)$ matrix. Lastly, $I$ is identity matrix and $L^+$ is Moore-Penrose Pseudoinverse of $L$.

**FGSD Definition**: For $x, y \in V$, we define the $f$-spectral distance between $x$ and $y$ on $G$ as follows:

$$\mathcal{S}_f(x, y) = \sum_{k=0}^{N-1} f(\lambda_k)(\phi_k(x) - \phi_k(y))^2 \tag{1}$$

We will refer to $\{\mathcal{S}_f(x, y)|f\}$, as the family of graph spectral distances. Without loss of generality, we assume that the derivative $f'(\lambda) \neq 0$ for $\lambda > 0$, and then by Lagrange Inversion Theorem [33], $f$ is invertible and thus *bijective*. For reasons that will be clear shortly, we are particularly interested in two sub-families of FGSD, where $f$ is monotonic function (increasing or decreasing) of $\lambda$. Depending on the sub-family, the $f$-spectral distance can capture different type of information in a graph.

**FGSD Elements Encode Local Structure Information**: For $f(\lambda) = \lambda^p$ $(p \geq 1)$, one can show that $\mathcal{S}_f(x, y) = (L^p)_{xx} + (L^p)_{yy} - 2(L^p)_{xy}$. If the shortest path from $x$ to $y$ is larger than $p$, then $(L^p)_{xy} = 0$. This is based on the fact $(L^p)_{xy}$ captures only $p$-hop local neighborhood information [32] on the graph. Hence, broadly for an increasing function of $f$ (e.g., a polynomial function of degree atleast $p \geq 1$), $\mathcal{S}_f(x, y)$ captures the *local* structure information.

**FGSD Elements Encode Global Structure Information**: On the other hand, $f$ as a decreasing function yields $\mathcal{S}_f(x, y) = ((L^+)^p)_{xx} + (((L^+)^p)_{yy} - 2((L^+)^p)_{xy}$. This captures the *global* information, since the $xy$-entry of $L^+ = (L + \frac{J}{N})^{-1} - \frac{J}{N}$ accounts for all paths from node $x$ to $y$ (and so does $(L^+)^p$). Several known globally aware graph distances can be derived from this FGSD sub-family. For $f(\lambda) = 1/\lambda$ where $\lambda > 0$, $\mathcal{S}_f(x, y)$ is the harmonic (or effective resistance) distance. More generally, for $f(\lambda) = 1/\lambda^p$, $p \geq 1$, $\mathcal{S}_f(x, y)$ is the polyharmonic distance ($p = 2$ is biharmonic distance). Lastly $f(\lambda_k) = e^{-2t\lambda_k}$ yields $\mathcal{S}_f(x, y)$ that is equivalent to the heat diffusion distance.

**FGSD Graph Signal Processing Point of View**: From graph signal processing perspective, $\mathcal{S}_f(x,y)$ is a distance computed based on spectral filter properties [32], where $f(\lambda)$ act as a band-pass filter. Or, it can be viewed in terms of spectral graph wavelets [15] as: $\mathcal{S}_f(x,y) = \psi_{f,x}(x) + \psi_{f,y}(y) - 2\psi_{f,x}(y)$, where $\psi_{f,x}(y) = \sum_{k=0}^{N-1} f(\lambda_k)\phi_k(x)\phi_k(y)$ (and $\psi_{f,x}(x), \psi_{f,y}(y)$ are similarly defined) is a spectral graph wavelet of scale 1, centered at node $x$ and $f(\lambda)$ act as a graph wavelet kernel.

**FGSD Based Graph Spectrum**: Using the FGSD based distance matrix $\mathcal{S}_f = [\mathcal{S}_f(x,y)]$ directly, e.g., for graph classification, requires us being able to solve the graph isomorphism problem efficiently. But no known polynomial time algorithm is available; the best algorithm today theoretically takes quasipolynomial time [2]. However, motivated from the study of homometric structure and the fact that each element of FGSD encodes some local or global sub-structure information of the graph, inspired us to define the graph spectrum as $\mathcal{R} = \{\mathcal{S}_f(x,y)|\forall(x,y) \in V\}$. Thus, comparing two $\mathcal{R}$'s *implicitly* evaluates the sub-structural similarity between two graphs. For instance, $\mathcal{R}$ based on harmonic distance contains sub-structural properties related to the *spanning trees* of a graph [29].

Our main concern in this paper would be choosing an appropriate $f(\lambda)$ function in order to generate $\mathcal{R}$ which can exhibit *ideal graph spectrum* properties as discuss below. Also, we want $\mathcal{F}$ to inherent these properties directly from $\mathcal{R}$, which is made possible by defining $\mathcal{F}$ as the histogram of $\mathcal{R}$. Finally, we lay down those important fundamental properties of an ideal graph spectrum that one would like $\mathcal{R}$ & $\mathcal{F}$ to obey on a graph $G = (V, E, W)$.

1. $\mathcal{R}$ & $\mathcal{F}$ must be invariant under any permutation $\pi$ of vertex labels. That is, $\mathcal{R}(G) = \mathcal{R}(G^\pi)$ or $\mathcal{R}(W) = \mathcal{R}(PWP^T)$ for any permutation matrix $P$.
2. $\mathcal{R}$ & $\mathcal{F}$ must have a unique representation for non-isomorphic graphs. That is, $\mathcal{R}(G_1) \neq \mathcal{R}(G_2)$ for any two non-isomorphic graphs $G_1$ and $G_2$.
3. $\mathcal{R}$ & $\mathcal{F}$ must be stable under small perturbation. That is, if graph $G_2(W_2) = G_1(W_1 + \Delta)$, for a small perturbation norm matrix $\|\Delta\|$, then the norm of $\|\mathcal{F}(G_2) - \mathcal{F}(G_1)\|$ should also be small or bounded in order to maintain the stability.
4. $\mathcal{F}$ must be sparse (if high-dimensional) for all the sparsity reasons desirable in machine learning.
5. $\mathcal{R}$ & $\mathcal{F}$ must be computationally fast for efficiency and scalability purposes.

## 4    Uniqueness of Family of Graph Spectral Distances and Graph Spectrum

We first start with exploring the graph invariance and uniqueness properties of $\mathcal{R}$ & $\mathcal{F}$ based on FGSD. Uniqueness is a very important (desirable) property, since it will determine whether the elements of $\mathcal{R}$ set are *complete* (i.e., how good they are), in the sense whether $\mathcal{R}$ is sufficient enough to recover all the intrinsic structural properties of a graph. We state the following important *uniqueness* theorem.

**Theorem 1 (Uniqueness of FGSD)** [3] *The $f$-spectral distance matrix $\mathcal{S}_f = [\mathcal{S}_f(x,y)]$ uniquely determines the underlying graph (up to graph isomorphism), and each graph has a unique $\mathcal{S}_f$ (up to permutation). More precisely, two undirected, weighted (and connected) graphs $G_1$ and $G_2$ have the same FGSD based distance matrix up to permutation, i.e., $\mathcal{S}_{G_1} = P\mathcal{S}_{G_2}P^T$ for some permutation matrix $P$, if and only if the two graphs are isomorphic.*

**Implications**: Our proof is based on establishing the following key relationship: $f(L) = -\frac{1}{2}(I - \frac{1}{N}J)\mathcal{S}_f(I - \frac{1}{N}J)$. Since $f$ is bijective, one can uniquely recover $\Lambda$ from $f(\Lambda)$. One of the consequence of Theorem 1 is that the $\mathcal{R}$ based on multiset of FGSD is invariant under the permutation of graph vertex labels and thus, satisfies the graph invariance property. Also, $\mathcal{F}$ will inherent this property since $\mathcal{R}$ remains the same. Unfortunately, it is possible that the multiset of some FGSD members can be same for non-isomorphic graphs (otherwise, we would have a $\mathcal{O}(N^2)$ polynomial time algorithm for solving graph isomorphism problem!). However, it is known that all non-isomorphic graphs with less than nine vertices have unique multisets of harmonic distance. While, for nine & ten vertex (simple) graphs, we have exactly 11 & 49 pairs of non-isomorphic graphs (out of total 274,668 & 12,005,168 graphs) with the same harmonic spectra. These examples show that there are *significantly very low* numbers of non-unique harmonic spectrums. Moreover, we empirically found that the biharmonic distance has all unique multisets for at-least upto ten vertices ($\sim$ 11 million graphs) and we couldn't find any non-isomorphic graphs with the same biharmonic multisets. Further, we have the following theorem regarding the uniqueness of $\mathcal{R}$.

**Theorem 2 (Uniqueness of Graph Harmonic Spectrum)** *Let $G = (V, E, W)$ be a graph of size $|V|$ with an unweighted adjacency matrix $W$. Then, if two graphs $G_1$ and $G_2$ have the same number of nodes but different number of edges, i.e, $|V_1| = |V_2|$ but $|E_1| \neq |E_2|$, then with respect to the harmonic distance multiset, $\mathcal{R}(G_1) \neq \mathcal{R}(G_2)$.*

**Implications**: Our proof relies on the fact that the effective resistance distance is a monotone function with respect to adding or removing edges. It shows that $\mathcal{R}$ based on some FGSD members specially harmonic distance is atleast theoretically known to be unique to a certain degree. $\mathcal{F}$ also inherent this property, fully under the condition $h \to 0$ (or for small enough $h$), where $h$ is the histogram binwidth.

Overall the certain uniqueness of $\mathcal{R}$ along with containing local or global structural properties in its each element dictate that the $\mathcal{R}$ is capable enough to serve as the *complete powerful* Graph Spectrum.

## 4.1 Unifying Relationship Between FSGD and Graph Embedding and Dimension Reduction

Before delving into other properties, we uncover an essential relationship between FGSD and Graph Embedding in Euclidean space and Dimension Reduction techniques. Let $f(\Lambda)^{\frac{1}{2}} = \mathbf{diag}[\sqrt{f(\lambda_k)}]$ and define $\boldsymbol{\Psi} = \boldsymbol{\Phi} f(\Lambda)^{\frac{1}{2}}$. Then, the $f$-spectral distance can be expressed as $\mathcal{S}_f(x, y) = ||\boldsymbol{\Psi}(x) - \boldsymbol{\Psi}(y)||_2^2$, where $\boldsymbol{\Psi}(x)$ is the $x^{th}$ row of $\boldsymbol{\Psi}$. Thus, $\boldsymbol{\Psi}$ represents an Euclidean embedding of $G$ where each node $x$ is represented by the vector $\boldsymbol{\Psi}(x)$. Now for instance, if $f(\lambda) = 1$, then by taking the first $p$ columns of $\boldsymbol{\Psi}$ yields embedding exactly equal to Laplacian Eigenmap (LE) [3] based on random walk graph Laplacian ($L_{rw} = D^{-1}L$). For $f(\lambda) = \lambda^{2t}$ and $L = D^{-1}W$, we get the Diffusion Map [25]. Thus, $f(\lambda)$ function has one-to-one correspondence relationship with spectral dimension reduction techniques. We have the following theorem concerning Graph Embedding based on FGSD.

**Theorem 3 (Uniqueness of FGSD Graph Embedding)** *Each graph $G$ can be isometrically embedded into a Euclidean space using FGSD as an isometric measure. This isometric embedding is* unique*, if all the eigenvalues of $G$ Laplacian are distinct and there does not exist any other graph $G'$ with Laplacian eigenvectors $\phi_k' = \sqrt{f(\lambda_j)/f(\lambda_j')}\phi_k, \forall k \in [1, N - 1]$.*

**Implications**: The above theorem shows that FGSD provides a unique way to embed the graph vertices into Euclidean space possibly without *loosing* any structural information of the graph. This could potentially serve as a cogent tool to convert an unstructured data into a structure data (similar to `structure2vec` 10 or `node2vec` 14 tool) which can enable us to perform standard inference tasks in Euclidean space. Note that the uniqueness condition is quite strict and holds for co-spectral graphs. In short, we have following uniqueness relationship, where $\boldsymbol{\Psi}$ is the Euclidean embedding of $G$ graph.

$$\mathcal{S}_f \underset{\longleftarrow}{\overset{\longrightarrow}{}} f(L_G) \longrightarrow L_G \longrightarrow f(L_G) \longrightarrow \boldsymbol{\Psi}_G$$

# 5 Stability of Family of Graph Spectral Distances and Graph Spectrum

Next, we hunt for the *stable members* of the FGSD that are robust against the perturbation or noise in the datasets. Specifically, we will look at the stability of $\mathcal{R}$ and $\mathcal{F}$ based on FGSD from $f(\lambda)$ perspective by first analyzing its influence on a single edge perturbation (or in other words analyzing rank one modification of Laplacian matrix). This will lead us to find the stable members and what restrictions we need to impose on $f(\lambda)$ function for stability. We will further show that $f$-spectral distance function also satisfies the notion of uniform stability [6] in a certain sense. For our analysis, we will restrict $f(\lambda)$ as a monotone function of $\lambda$, for $\lambda > 0$. Let $\triangle w \geq 0$ be the change after modifying $w$ weight on any single edge to $w'$ on the graph, where $\triangle w = w' - w$.

**Theorem 4 (Eigenfunction Stability of FGSD)** *Let $\triangle\mathcal{S}_{xy}$ be the change in $\mathcal{S}_f(x, y)$ distance with respect to $\triangle w$ change in weight of any single edge on the graph. Then, $\triangle\mathcal{S}_{xy}$ for any vertex pair $(x, y)$ is bounded with respect to the function of eigenvalue as follows,*

$$\triangle\mathcal{S}_{xy} \leq 2\big(|f(\lambda_{N-1} + 2\triangle w) - f(\lambda_1)|\big)$$

**Implications**: Since, $\mathcal{R} = \{\mathcal{S}_f(x, y)|\forall(x, y) \in V\}$, then each element of $\mathcal{R}$ is itself bounded by $\triangle\mathcal{S}_{xy}$. Now, recall that $\mathcal{F}$ is a histogram of $\mathcal{R}$, then $\mathcal{F}$ won't change, if binwdith is large enough to accommodate the perturbation i.e., $h \geq 2\triangle\mathcal{S}_{xy} \forall(x, y)$ assuming all elements of $\mathcal{R}$ are

at the center of their respective histogram bins. Besides $h$, the other way to make $\mathcal{R}$ *robust* is by choosing a suitable $f(\lambda)$ function. Lets consider the behavior $\triangle \mathcal{S}_{xy}$ on $f(\lambda) = \lambda^p$ for $p > 0$. Then, $\triangle \mathcal{S}_{xy} \leq 2\big((\lambda_{N-1} + 2\triangle w)^p - \lambda_1^p\big)$ and as a result, $\triangle \mathcal{S}_{xy}$ is an increasing function with respect to $p$ which implies that stability decreases with increase in $p$. For $p = 0$, stability does not change with respect to $\lambda$. While, for $p < 0$, $\triangle \mathcal{S}_{xy} \leq 2\big(1/\lambda_1^{|p|} - 1/(\lambda_{N-1} + 2\triangle w)^{|p|}\big)$. Here, $\triangle \mathcal{S}_{xy}$ is a decreasing function with respect to $|p|$, which implies that stability increases with decrease in $p$. The results conforms with the reasoning that eigenvectors corresponding to smaller eigenvalues are smoother (i.e., oscillates slowly) than large eigenvectors (corresponding to large eigenvalues) and decreasing $p$ will attenuate the contribution of large eigenvectors, making the $f$-spectral distance more stable and less susceptible towards perturbation or noise. However, decreasing $p$ too much could result in lost of local information contained in eigenvectors with larger eigenvalues and therefore, a balance needs to be maintained. Overall, Theorem 4 shows that either through suitable $h$ or decreasing $f(\lambda)$ function, stability of $\mathcal{R}$ & $\mathcal{F}$ can be controlled to satisfy the Ideal Spectrum Property 3.

Infact, we can further show that $\mathcal{S}_f(x, y)$ between any two vertex $(x, y)$ on a graph, with $0 < \alpha \leq w \leq \beta$ bounded weights, is tightly bounded to a certain expected value.

**Theorem 5 (Uniform Stability of FGSD)** *Let $\mathbf{E}[\mathcal{S}_f(x, y)]$ be the expected value of $\mathcal{S}_f(x, y)$ between vertex pair $(x, y)$, over all possible graphs with fixed ordering of $N$ vertices. Then we have, with probability $1 - \delta$, where $\delta \in (0, 1)$ and $\theta$ depends upon $\alpha, \beta, N$.*

$$\big|\mathcal{S}_f(x, y) - \mathbf{E}[\mathcal{S}_f(x, y)]\big| \leq f(\theta)\sqrt{N(N-1)}\sqrt{\log \frac{1}{\delta}}$$

**Implications**: The above theorem is based on the fact $\triangle \mathcal{S}_{xy}$ can itself be upper bounded over all possible graphs generated on a fixed ordering of $N$ vertices. This is a very similar condition needed for a learning algorithm to satisfy the notion of uniform stability in order to give generalization guarantees. The $f$-spectral distance function can itself be thought of as a learning algorithm which admits uniform stability (precise definition in supplementary) and indicates a strong stability behavior over all possible graphs and further act as a generalizable learning algorithm on the graph. Theorem 5 also reveals that the deviation can be minimized by choosing decreasing $f(\lambda)$ function and it would be suitable, if $f(\lambda)$ grow with $\mathcal{O}\big(1/\sqrt{N(N-1)}\big)$ rate in order to maintain stability for large graphs.

So far, we have narrow down our interest to $\mathcal{R}$ & $\mathcal{F}$ based on the bijective and decreasing $f(\lambda)$ function for achieving both uniqueness and stability. This eliminates all forms of increasing polynomial functions as a good choice of $f(\lambda)$. As a result, we can focus on inverse (or rational) form of polynomial functions such as polyharmonic distances. A by-product of our analysis results in revealing a new class of stable dimension reduction techniques, possible by scaling Laplacian eigenvectors with decreasing function of $f(\lambda)$, although such connections have already been known before.

## 6 Sparsity of Family of Graph Spectral Distances and Graph Spectrum

Figure 2: Figure shows the number of unique elements present in $\mathcal{R}$ formed by different $f$-spectral distance on all graphs (of $|V| = 9$, total $261,080$ graphs). Graph enumeration indices are sorted according to $\big|\mathcal{R}(\frac{1}{\lambda})\big|_G$. We can observe that $f(\lambda) = \frac{1}{\lambda}$ increases in form of a step function and lower bounds all other $f(\lambda)$ with an addition constant. (Best viewed in color and when zoom-in.)

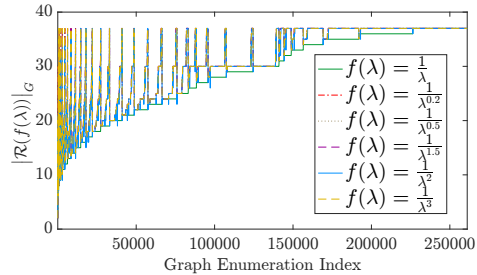

Sparsity is desirable for both computational and statistical efficiency. In this section, we investigate the sparsity produced in $\mathcal{F}$ by choosing different $f(\lambda)$ functions. Here, sparsity refers to its usual definition of "how many zero features are present in $\mathcal{F}$ graph feature vector". Since $\mathcal{F}$ is a histogram of $\mathcal{R}$, number of non-zero elements in $\mathcal{F}$ will always be less than equal to number of unique (or distinct) elements in $\mathcal{R}$. However, due to the lack of any theoretical support, we rely on empirical evidence and conjecture the following statement.

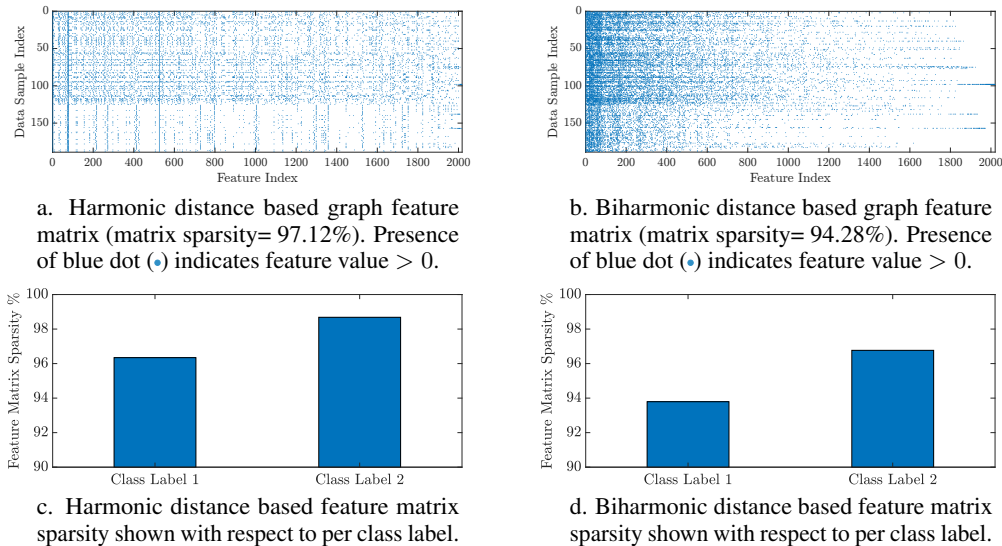

a. Harmonic distance based graph feature matrix (matrix sparsity= 97.12%). Presence of blue dot (•) indicates feature value $> 0$.

b. Biharmonic distance based graph feature matrix (matrix sparsity= 94.28%). Presence of blue dot (•) indicates feature value $> 0$.

c. Harmonic distance based feature matrix sparsity shown with respect to per class label.

d. Biharmonic distance based feature matrix sparsity shown with respect to per class label.

Figure 3: Feature space for MUTAG (composed of two class sizes 125 & 63): Both harmonic & biharmonic based graph spectrum encodes a sparse high dimensional feature representation $\mathcal{F}$ for graphs which can clearly *distinguish* the two classes as depicted in above sub-figures.

**Conjecture (Sparsity of FGSD Graph Spectrum)** *For any graph $G$, let $\left|\mathcal{R}(f(\lambda))\right|_G$ represents the number of unique elements present in the multiset of $\mathcal{R}$, computed on an unweighted graph $G$ based on some monotonic decreasing $f(\lambda)$ function. Then, the following holds,*

$$\left|\mathcal{R}(f(\lambda))\right|_G \geq \left|\mathcal{R}\left(\frac{1}{\lambda}\right)\right|_G + 2$$

The conjecture is based on the observation that, in the Figure 2, $\left(\left|\mathcal{R}\left(\frac{1}{\lambda}\right)\right| + 2\right)$ lower bounds all given monotonic decreasing $f(\lambda)$ along with an addition constant of 2. Same trends are observed for different graph sizes $|V|$. Interestingly, when graph enumeration indices are sorted according to size $\left|\mathcal{R}\left(\frac{1}{\lambda}\right)\right|$, we further observe that $f(\lambda) = \frac{1}{\lambda}$ increases in the form of a step function. From this conjecture, we can directly conclude that the $\mathcal{F}$ based on $f(\lambda) = \frac{1}{\lambda}$ produce the most sparse features because number of unique elements in its $\mathcal{R}$ is always less than any other $\mathcal{R}$. Figure 3, further supports this conjecture which shows the feature space computed for MUTAG dataset in case of harmonic and biharmonic spectrums. However, this raises a question of trade-off between maintaining uniqueness and sparsity, since biharmonic distance multisets are found to be unique for more number of graphs than harmonic distance. Nonetheless, some preliminary experiments measuring harmonic vs. biharmonic performance on graph classification (in supplementary), suggest that the sparsity is more favorable than uniqueness since it results in higher classification accuracy.

## 7 Fast Computation of Family of Graph Spectral Distances and Spectrum

Finally, we provide the general recipe of computing any member of FGSD in fast manner. In order to avoid direct eigenvalue decomposition, we can either perform approximation or leverage structural properties and sparsity of $f(L)$ for efficient exact computation of $\mathcal{S}_f$ and thus, $\mathcal{R}$.

**Approximation**: Inspired from the spectral graph wavelet work [32], the recipe for approximating FGSD is to decompose $f(\lambda)$ possibly into an approximate polynomial series (for example, chebyshev polynomials) as follows: $f(\lambda) = \sum_{i=0}^{r} a_i T_i(\lambda)$ such that $T_i(x)$ can be computed in recursive manner from few lower order terms $(T_{i-1}(x), T_{i-2}(x), ..., T_{i-c}(x))$. Then it follows, $f(L) = \sum_{i=0}^{r} a_i T_i(L)$. In this case, the cost of computing will reduce to $\mathcal{O}(r|E|)$ for sparse $L$ which is very less expensive, since $\mathcal{O}(r|E|) \ll \mathcal{O}(N^2)$. But, if $f(\lambda)$ is an inverse polynomial form of function, then computing $f(L) = \left(\sum_{i=0}^{r} a_i T_i(L)\right)^{-1} = f(L_r^+)$, boils down to efficiently computing (a single) Moore Penrose Pseudo inverse of a matrix.

**Efficient Exact Computation**: By leveraging $f(L)$ structural properties and its sparsity, we can efficiently perform exact computation of $f(L^+)$ in much more better way than the eigenvalue

decomposition. We *propose* such a **method** 1 which is the generalization of [23] work. We can show that, $f(L)f(L^+)^{-1} = I - \frac{J}{N}$. Therefore, $f(L)l_k^+ = B_k$, where $l_k^+$ and $B_k$ are the $k^{th}$ column of $f(L^+)$ and $B = I - \frac{J}{N}$ matrices, respectively. So, first we can find a particular solution of following (sparse) linear system: $f(L)\mathbf{x} = B_k$ and then obtain $l_k^+ = \mathbf{x} - \frac{\mathbf{1}^T\mathbf{x}}{\mathbf{1}^T\mathbf{1}}\mathbf{x}$. The particular solution $\mathbf{x}$ can be obtained by replacing any single row and corresponding column of $f(L)$ by zeros, and setting diagonal entry at their intersection to one, and replacing corresponding row of $B$ by zeros. This gives a (non-singular) sparse linear system which can be solved very efficiently by performing cholesky factorization and back-substitution, resulting in overall $\mathcal{O}(N^2)$ complexity as shown in [5]. Beside this, there are few other fast methods to compute Pseudo inverse, particularly given by [17].

| Complexity | SP [4] | GK[34]($k \in \{3,4,5\}$) ($d \leq N$) | SGS[20] | GS [22]($k \in [2,6]$) | DCNN[1] | MLG[21] ($\widetilde{N} < N$) | **FGSD** |
|---|---|---|---|---|---|---|---|
| Approximate | — | $\mathcal{O}(Nd^{k-1})$ | — | — | — | $\mathcal{O}(\widetilde{N}^3)$ | $\mathcal{O}(r|E|)$ |
| Worst-Case | $\mathcal{O}(N^3)$ | $\mathcal{O}(N^k)$ | $\mathcal{O}(N^3)$ | $\mathcal{O}(N^{2+k})$ | $\mathcal{O}(N^2)$ | $\mathcal{O}(\widetilde{N}^3)$ | $\mathcal{O}(N^2)$ |

Table 1: FGSD complexity comparison with few strong state-of-art algorithms (showing variables that are only dependent on $N$ & $|E|$). It reveals that the FGSD complexity is better than the most.

As a result, it leads to a *very efficient* $\mathcal{O}(r|E|)$ complexity through approximation with the *worst-case* $\mathcal{O}(N^2)$ complexity in exact computation of $\mathcal{R}$. Table 1, shows the complexity comparison with other state-of-art methods. Since, number of elements in $\mathcal{R}$ are $\mathcal{O}(N^2)$, then $\mathcal{F}$ is also bounded by $\mathcal{O}(N^2)$ and thus satisfies the ideal graph spectrum Property 5. Finally, Algorithm 1 summarizes the complete procedure of computing $\mathcal{R}$ & $\mathcal{F}$.

---
**Algorithm 1** Computing $\mathcal{R}$ and $\mathcal{F}$ based on FGSD.

---
**Input:** Given graphs $\{G_i = (V_i, E_i, W_i)\}_{i=1}^M$, $f(\lambda)$, number of bins $b$, binwidth $h$.
**Output:** $\mathcal{R}_i$ and $\mathcal{F}_i \forall i \in [1, M]$.
    **for** $i = 1$ to $M$ **do**
        Compute $f(L_i)$ using approx. or exact method 1.
        Compute $\mathcal{S}_i = \mathbf{diag}(f(L_i))J + J\mathbf{diag}(f(L_i)) - 2f(L_i)$.
        Set $\mathcal{R}_i = \{\mathcal{S}_{xy}|\forall(x,y) \in |V_i|\}$.
        Compute $\mathcal{F}_i = histogram(\mathcal{R}_i, b, h)$.
    **end for**

---

# 8 Experiments and Results

**FGSD Graph Spectrum Settings**: We chose harmonic distance as an ideal candidate for $\mathcal{F}$. For fast computation, we adopted our proposed efficient exact computation method 1. And for computing histogram, we fix binwidth size and set the number of bins such that its range covers all $\{\mathcal{R}_i\}_1^M$ elements of $M$ number of graphs. Therefore, we had only one parameter, binwidth size, chosen from the set $\{0.001, 0.0001, 0.00001\}$. This results in $\mathcal{F}$ feature vector dimension in range $100 - 1000,000$ with feature matrix sparsity $> 90\%$ in all cases. Our FGSD code is available at github[4].

**Datasets**: We employed wide variety of datasets considered as benchmark [1, 34, 21, 26] in graph classification task to evaluate the quality of produce FGSD graph features. We adopted **7 bioinformatics** datasets: Mutag, PTC, Proteins, NCI1, NCI109, D&D, MAO and **5 social network** datasets: Collab, REDDIT-Binary, REDDIT-Multi-5K, IMDB-Binary, IMDB-Multi. D&D dataset contains 691 enzymes and 587 non-enzymes proteins structures. While, MAO dataset contains 38 molecules that are antidepressant drugs and 30 do not. For other datasets, details can be found in [34].

**Experimental Set-up**: All experiments were performed on a single Intel-Core i7-4790@3.60GHz and 64GB RAM machine. We compare our method with **6 state-of-art Graphs Kernels**: Random Walk (RW) [13], Shortest Path Kernel (SP) [4], Graphlet Kernel (GK) [30], Weisfeiler-Lehman Kernel (WL) [31], Deep Graph Kernels (DGK) [34], Multiscale Laplacian Graph Kernels (MLK) [21]. And proposed, **2 recent state-of-art Graph Convolutional Networks**: PATCHY-SAN (PSCN) [26], Diffusion CNNs (DCNN) [1]. And, **2 strong Graph Spectrums**: the Skew Spectrum (SGS) [20], Graphlet Spectrum (GS) [22]. We adopt the same procedure from previous works [26, 34] to make a fair comparison and used 10-fold cross validation with LIBSVM [8] library to test the classification performance. Parameters of SVM are independently tuned using training folds data and best average classification accuracies is reported for each method. We provide node degree as the labeled data for algorithms that do not operate directly on unlabeled data. Further details about parameters selection for baseline methods are present in supplementary materials.

| Dataset (No. Graphs, Max. Nodes) | RW [2003] | SP [2005] | GK [2009] | WL [2011] | DGK [2015] | MLG (Wall-Time) [2016] | DCNN [2016] | SGS [2008] | **FGSD** (Wall-Time) |
|---|---|---|---|---|---|---|---|---|---|
| MUTAG (188, 28) | 83.50 | 87.23 | 84.04 | 87.28 | 86.17 | 87.23(5$s$) | 66.51 | 88.61 | **92.12**(**0.3s**) |
| PTC (344, 109) | 55.52 | 58.72 | 60.17 | 55.61 | 59.88 | **62.20**(18$s$) | 55.79 | — | **62.80**(**0.07s**) |
| PROTEINS (1113, 620) | 68.46 | 72.14 | 71.78 | 70.06 | 71.69 | 71.35(277$s$) | 65.22 | — | **73.42**(**5s**) |
| NCI1 (4110, 111) | > D | 68.15 | 62.07 | 77.23 | 64.40 | 77.57(620$s$) | 63.10 | 62.72 | **79.80**(**31s**) |
| NCI109 (4127, 111) | > D | 68.30 | 62.04 | **78.43** | 67.14 | 75.91(600$s$) | 60.67 | 62.62 | **78.84**(**35s**) |
| D & D (1178, 5748) | > D | > D | 75.05 | 73.76 | 72.75 | **77.02**(7.5$hr$) | OMR | — | **77.10**(**25s**) |
| MAO (68, 27) | 83.52 | 90.35 | 80.88 | 89.79 | 87.76 | 91.17(13$s$) | 76.10 | — | **95.58**(**0.1s**) |

Table 2: Classification accuracy on *unlabeled* bioinformatics datasets. Results in **bold** indicate all methods with accuracy within range 2.0 from the top result and **blue color** (for range > 2.0), indicates the new state-of-art result. **Green color** highlights the best time computation, if it's **5×faster** (among the mentioned). 'OMR' is out of memory error, '> D' is computation exceed 24$hrs$.

| Dataset (Graphs) | GK [2009] | DGK [2015] | PSCN [2016] | **FGSD** |
|---|---|---|---|---|
| COLLAB (5000) | 72.84 | 73.09 | 72.60 | **80.02** |
| REDDIT-B (2000) | 77.34 | 78.04 | **86.30** | **86.50** |
| REDDIT-M (5000) | 41.01 | 41.27 | **49.10** | **47.76** |
| IMDB-B (1000) | 65.87 | 66.96 | 71.00 | **73.62** |
| IMDB-M (1500) | 43.89 | 44.55 | 45.23 | **52.41** |

Table 3: Classification accuracy on social network datasets. FGSD significantly outperforms other methods.

| Dataset | MLG [2016] | DCNN [2016] | PSCN [2016] | GS [2009] | **FGSD\*** |
|---|---|---|---|---|---|
| MUTAG | 87.94 (4$s$) | 66.98 | **92.63** (3$s$) | 88.11 | **92.12** (**0.3s**) |
| PTC | **63.26** (21$s$) | 56.60 | **62.90** (6$s$) | — | **62.80** (**0.07s**) |
| NCI1 | **81.75** (621$s$) | 62.61 | **78.59** (76$s$) | 65.0 | **79.80** (31$s$) |
| D & D | **78.18** (7.5$hr$) | OMR | **77.12** (154$s$) | — | **77.10** (**25s**) |
| MAO | 88.29 (12$s$) | 75.14 | — | — | **95.58** (**0.1s**) |

Table 4: Classification accuracy on *labeled* bioinformatics datasets. **\*** emphasize that FGSD did *not* utilize any node labels.

.

**Classification Results**: From Table 2, it is clear that FGSD consistently **outperforms** every other state-of-art algorithms on unlabeled bioinformatics datasets and that too *significantly* in many cases. FGSD even performs better for social network graphs as shown in Table 3 and achieves a *very significant* **7% − 8%** more accuracy than the current state-of-art PSCNs on COLLAB and IMDB-M datasets. Also from run-time perspective (excluding any data loading or classification time for all algorithms), it is **pretty fast** (2x–1000x times faster) as compare to others. These appealing results further motivated us to compare FGSD on the labeled datasets (*even though, it is not a complete fair comparison*). Table 4 shows that FGSD is still **very competitive** with all other strong (recent) algorithms that utilize node labeled data. Infact on MAO dataset, FGSD sets a *new* state-of-art result and stays *within* **0% − 2%** range of accuracy from the best on all labeled datasets. On few labeled datasets, we found MLG to have slightly better performance than the others, but it is 1000 *times slower* than FGSD when graph size jumps to few thousand nodes (see D&D Results). Altogether, FGSD shows very promising results in both accuracy & speed on all type of datasets and over all the more sophisticated algorithms. These results also point out the fact that there is untapped hidden potential in the graph structure which current algorithms are not harnessing despite having labeled data at their disposal.

## 9 Conclusion

We present a conceptually simple yet powerful and theoretically motivated graph representation. In particular, our graph representation based on the discovery of family of graph spectral distances can exhibits uniqueness, stability, sparsity and are computationally fast. Moreover, our hunt specifically leads to the harmonic and next to it, biharmonic distances as an ideal members of this family for extracting graph features. Finally, our extensive results show that FGSD based graph features are powerful enough to dominate the unlabeled graph classification task over all the more sophisticated algorithms and competitive enough to yield high classification accuracy on labeled data even without utilizing any node labels. **In our future work**, we plan to *generalize* the FGSD for labeled dataset in order to utilize the useful node and edge label information in the graph representation.

## 10    Acknowledgments

This research was supported in part by ARO MURI Award W911NF-12-1-0385, DTRA grant HDTRA1-14-1-0040, and NSF grants CNS-1618339, CNS-1618339 and CNS-1617729.

## Footnotes

[1]That is, invariant under permutation of graph vertex labels.

[2]A set in which an element can occur multiple of times.

[3]Variant of Theorem 1 also hold true for the normalized graph Laplacian $L_{norm} = D^{-\frac{1}{2}}LD^{-\frac{1}{2}}$.

[4]https://github.com/vermaMachineLearning/FGSD

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
