[Supplementary Material]

# Supplementary Material: Hunt For The Unique, Stable, Sparse And Fast Feature Learning On Graphs

**Saurabh Verma**
Department of Computer Science
University of Minnesota, Twin Cities
verma@cs.umn.edu

**Zhi-Li Zhang**
Department of Computer Science
University of Minnesota, Twin Cities
zhang@cs.umn.edu

## 1   Proof of Theorem 1

**Notations**: Let $G = (V, E, W)$ be an undirected graph with vertex set $V$, edge set $E$, and $W$ as a (symmetric) nonnegative weighted adjacency (affinity) matrix. In particular, $W_{ij} = W(i, j) > 0$ if and only if $(i, j) \in E$. We will assume $G$ is connected for all proofs unless mentioned otherwise. The (standard) graph Laplacian is defined as $L = D - W$, where $D = \mathbf{diag}[d_1, .., d_n]$ and $d_i = d(i) = \sum_{j \neq i}^{n} W(i, j)$ the (weighted) degree of node $i$. $L$ is semi-definite and admits an eigenvalue decomposition of the form $L = \mathbf{\Phi} \Lambda \mathbf{\Phi}^T$, where $\Lambda = \mathbf{diag}[\lambda_k]$ is the diagonal matrix formed by the eigenvalues $\lambda_0 = 0 < \lambda_1 \leq \cdots \leq \lambda_{N-1}$, and $\mathbf{\Phi} = [\phi_0, ..., \phi_{N-1}]$ is an orthogonal matrix formed by the corresponding eigenvectors $\phi_k$'s. For $x \in V$, we use $\phi_k(x)$ to denote the $x$-entry value of $\phi_k$. Let $f$ be an arbitrary nonnegative (real-analytical) function on $R^+$ with $f(0) = 0$. Then $f(L)$ is defined as $f(L) = \mathbf{\Phi} f(\Lambda) \mathbf{\Phi}^T$. In addition, let $\mathbf{1} = [1, 1, ..., 1]^T = \sqrt{N} \phi_0$ be all-one column vector, $J = \mathbf{1}\mathbf{1}^T = N \phi_0 \phi_0^T$. For any matrix $M$, we use $\mathbf{diag}(M)$ to denote a diagonal matrix consisting of the diagonal entries of $M$, namely, $\mathbf{diag}(M) = \mathbf{diag}[M_{00}, M_{11}, \ldots, M_{N-1,N-1}]$. For conciseness, we will drop subscript $f$ when context is clear.

**Proof**: We first note that the graph $f$-spectral distance matrix $\mathcal{S}_f = [\mathcal{S}_f(x, y)]$, can be expressed in the matrix form using $f(L)$ as follows:

$$
\begin{aligned}
\mathcal{S}_f &= \mathbf{diag}(\mathbf{\Phi} f(\Lambda) \mathbf{\Phi}^T) J + J \, \mathbf{diag}(\mathbf{\Phi} f(\Lambda) \mathbf{\Phi}^T) - 2 \mathbf{\Phi} f(\Lambda) \mathbf{\Phi}^T \\
&= \mathbf{diag}(f(L)) J + J \, \mathbf{diag}(f(L)) - 2 f(L)
\end{aligned}
\tag{1}
$$

Assume, graphs $G_1$ and $G_2$ are isomorphic and do not contain self-loops then their respective adjacency matrices are equal upto permutation ($W_1 = P W_2 P^T$) and also implies the equality of their Laplacian matrices upto permutation ($L_1 = P L_2 P^T$). Then, we can specify $\mathcal{S}_{G_1}$ in terms of $\mathcal{S}_{G_2}$ as follows,

$$
\begin{aligned}
P \mathcal{S}_{G_2} P^T &= P \mathbf{diag}(f(L_2)) J P^T + P J \, \mathbf{diag}(f(L_2)) P^T - 2 P f(L_2) P^T \\
&= \mathbf{diag}(P f(L_2) P^T) \mathbf{1}^T + J \, \mathbf{diag}(P f(L_2) P^T) - 2 P f(L_2) P^T
\end{aligned}
\tag{2}
$$

Since $f$ only operates on eigenvalues of a matrix and $P$ can only permutes the rows of $\phi$ matrix, we have $f(L_1) = P f(L_2) P^T$. This *proves* that $\mathcal{S}_{G_1} = P \mathcal{S}_{G_2} P^T$ for any *two isomorphic graphs*. Above result, also holds for $L_{norm}$. Now, all it remains to show that this also holds true in opposite direction as well. More specifically, we prove that one can uniquely recover Laplacian matrix $L$ from the $f$-spectral distance matrix $\mathcal{S}_f$ as follows:

Since, the eigenvectors of symmetric $L$ are orthogonal to each other and to $\mathbf{1}$, we have,

$$
f(L)\mathbf{1} = \sum_{k=0}^{N-1} f(\lambda_k) \mathbf{u}_k \mathbf{u}_k^T \mathbf{1} = \mathbf{0}, \text{ since } f(\lambda_0) = 0
\tag{3}
$$

Similarly, we have $\mathbf{1}^T f(L) = 0$. Using Eq. 1 and 3, we can explicitly derive $f(L)$ in terms of $D$ as,

$$f(L) = -\frac{1}{2}\Big(\mathcal{S}_f - \frac{1}{N}(\mathcal{S}_f J + J\mathcal{S}_f) + \frac{1}{N^2}J\mathcal{S}_f J\Big) \tag{4}$$

This shows that we can uniquely recover $f(L)$ from $\mathcal{S}_f$. Further since $f(\lambda)$ is a bijective function, we can also uniquely recover the eigenvector matrix and eigenvalues of $L$ from $f(L)$.

Finally, we show that for a given two graph $f$-spectral distance matrices, $\mathcal{S}_1 = \mathcal{S}_{G_1}$ and $\mathcal{S}_2 = \mathcal{S}_{G_2}$, such that $\mathcal{S}_2 = P\mathcal{S}_1 P^T$ for some permutation matrix, we have $L_2 = PL_1 P^T$.

$$\begin{aligned} Pf(L_1)P^T &= -\frac{1}{2}P\Big(\mathcal{S}_1 - \frac{1}{N}(\mathcal{S}_1 J + J\mathcal{S}_1) + \frac{1}{N^2}J\mathcal{S}_1 J\Big)P^T \\ &= -\frac{1}{2}\Big(P\mathcal{S}_1 P^T - \frac{1}{N}(P\mathcal{S}_1 P^T J + JP\mathcal{S}_1 P^T) + \frac{1}{N^2}JP\mathcal{S}_1 P^T J\Big) \\ &= -\frac{1}{2}\Big(\mathcal{S}_2 - \frac{1}{N}(\mathcal{S}_2 J + J\mathcal{S}_2) + \frac{1}{N^2}J\mathcal{S}_2 J\Big) \\ &= f(L_2) \end{aligned}$$

Since $f$ is bijective, we can conclude $Pf(L_1)P^T = f(L_2) \implies PL_1 P^T = L_2$. *This completes the full proof of Theorem 1.*

**Variant of Theorem 1**: We consider the variant of Theorem 1 for the *normalized* graph Laplacian $L_{norm} = \tilde{L} = D^{-\frac{1}{2}}LD^{-\frac{1}{2}}$. Again $\tilde{L} = D^{-\frac{1}{2}}LD^{-\frac{1}{2}}$ is semi-definite and admits an eigendecomposition of the form $\tilde{L} = \tilde{\mathbf{\Phi}}\tilde{\Lambda}\tilde{\mathbf{\Phi}}^T$, where $\tilde{\Lambda} = \mathbf{diag}[\tilde{\lambda}_k]$ is the diagonal matrix formed by the eigenvalues $\tilde{\lambda}_0 = 0 < \tilde{\lambda}_1 \leq \cdots \leq \tilde{\lambda}_{N-1}$, and $\tilde{\mathbf{\Phi}} = [\tilde{\phi}_0, ..., \tilde{\phi}_{N-1}]$ is an orthogonal matrix formed by the corresponding eigenvectors $\tilde{\phi}_k$'s.

With respect to $\tilde{L}$, for a given (real)-analytical function $f$ on $R^+$ with $f(0) = 0$, we define the $f$-spectral distance between $x$ and $y$ on $G$ as follows:

$$\tilde{\mathcal{S}}_f(x,y) = \sum_{k=0}^{N-1} f(\tilde{\lambda}_k)(\tilde{\phi}_k(x) - \tilde{\phi}_k(y))^2 \tag{5}$$

Define, $\tilde{J} = D^{\frac{1}{2}}J$, $d = \text{trace}(D^{\frac{1}{2}})$ and $\text{trace}(L) = \sum_i \sum_j W_{ij} = vol(G)$ (the sum of the weights of all edges). We note that $\tilde{\phi}_k^T D^{\frac{1}{2}}\mathbf{1} = 0$, $\forall k \neq 0$, and thus, $\tilde{L}^T\tilde{J} = \tilde{J}^T\tilde{L} = 0$. Corresponding to Eq. 1 and Eq. 4, the following expressions hold which relate the (normalized) graph $f$-spectral distance matrix $\tilde{\mathcal{S}}_f = [\tilde{\mathcal{S}}_f(x,y)]$ to the normalized graph Laplacian $\tilde{L}$,

$$\tilde{\mathcal{S}}_f = \mathbf{diag}(f(\tilde{L}))J + J\,\mathbf{diag}(f(\tilde{L})) - 2f(\tilde{L})$$

and

$$f(\tilde{L}) = -\frac{1}{2}\Big(\tilde{\mathcal{S}}_f - \frac{1}{d}(\tilde{\mathcal{S}}_f\tilde{J} + \tilde{J}\tilde{\mathcal{S}}_f) + \frac{1}{d^2}\tilde{J}\tilde{\mathcal{S}}_f\tilde{J}\Big) \tag{6}$$

Hence, we have the following variant of Theorem 1.

**Theorem 1 (Uniqueness of Normalized FGSD)** *The $f$-spectral distance matrix $\mathcal{S}_f = [\mathcal{S}_f(x,y)]$ uniquely determines the underlying graph (up to graph isomorphism and a scalar constant factor $d$ on the weigh matrix). Thus, each graph has a unique $\mathcal{S}_f$ up to permutation. More precisely, two undirected, weighted (and connected) graphs $G_1$ and $G_2$ have the same FGSD based distance matrix up to permutation, i.e., $\mathcal{S}_{G_1} = P\mathcal{S}_{G_2}P^T$ for some permutation matrix $P$, if and only if the two graphs are isomorphic and the weight matrices $W_{G_1}$ and $W_{G_2}$ is such that $W_{G_1} = PdW_{G_2}P^T$ for some scalar constant $d > 0$.*

## 2 Proof of Theorem 2

The proof directly builds upon the fact that the effective resistance distance on the graph is a monotone function with respect to adding or removing edges (or weights) (see in [4], Theorem 2.6 for more details). In other words, pairwise effective resistance on a graph cannot increase when edges (or weights) are added. Suppose we add an edge (or increase the edge weight) $w_{ij}$ on a graph $G$ and get a new graph $G^{'}$. Let $\mathcal{R}$ and $\mathcal{R}^{'}$ be the multiset of pairwise effective resistance distances before and after adding/increasing the edge (weight) $w_{ij}$ in the graph respectively.

Then, using the parallel law of resistance, we can show that $\mathcal{S}^{'}_{ij}$ will always decrease according to $\mathcal{S}^{'}_{ij} = \frac{\mathcal{S}_{ij} w_{ij}}{\mathcal{S}_{ij} + w_{ij}}$. As a result, if the elements of $\mathcal{R}$ and $\mathcal{R}^{'}$ are in the sorted order, then the following vector comparison (element-wise) strictly holds: $\mathcal{R}^{'} \prec \mathcal{R}$, when we add edges. Likewise, $\mathcal{R}^{'} \succ \mathcal{R}$ when we remove edges. Thus, we conclude that $\mathcal{R}$ is unique up to a fixed number of edges.

## 3 Proof of Theorem 3

FGSD provides a graph embedding using the following expression, $\mathbf{\Psi} = \mathbf{\Phi}\sqrt{f(\Lambda)}$. Hence, we have $\mathcal{S}_f(x, y) = ||\mathbf{\Psi}(x) - \mathbf{\Psi}(y)||^2_2$. By the definition of graph isometric embedding, this shows that FGSD serves as an isometric measure on the graph embedding (defined by $\mathbf{\Psi}$) in a Euclidean space.

Next, we show the conditions under which $\mathbf{\Psi}$ embedding is unique for a graph $G$.

**1)** $\mathbf{\Psi}$ is unique, if there does not exist any other $\mathbf{\Psi}' = \mathbf{\Psi}$ with $f(L^{'}) \neq f(L^{'})$. As a result, $\mathbf{\Phi}^{'}\sqrt{f(\Lambda^{'})} \neq \mathbf{\Phi}\sqrt{f(\Lambda)}$ must hold true for all $L^{'} \neq L$. This implies that, $\phi^{'}_k \neq \sqrt{f(\lambda_j)/f(\lambda^{'}_j)}\phi_k$, $\forall k \in [1, N-1]$.

**2)** We must also make sure that one can always reconstruct the same $\mathbf{\Psi}$ from $\mathcal{S}_f$. Now, according to Theorem 1, one can recover $f(L)$ uniquely from $\mathcal{S}_f$. If all the eigenvalues of $f(L)$ are distinct, then the eigenvalue decomposition of $L$ is also unique. This ensures that we can recover the same $\mathbf{\Phi}$ after the decomposition to reconstruct the same $\mathbf{\Psi}$. Note that we can compute $\mathcal{S}_f$ directly from $\mathbf{\Psi}$ as follows: $\mathcal{S}_f = \mathbf{diag}(\mathbf{\Psi}\mathbf{\Psi}^T)J + J\,\mathbf{diag}(\mathbf{\Psi}\mathbf{\Psi}^T) - 2\mathbf{\Psi}\mathbf{\Psi}^T$. In short, we have the following uniqueness relationships under the aforementioned conditions, where $\mathbf{\Psi}$ is the Euclidean embedding of a graph $G_1$:

$$\mathcal{S}_{G_1} \underset{\longrightarrow}{\rightleftarrows} f(L_{G_1}) \longrightarrow L_{G_1} \longrightarrow f(L_{G_1}) \longrightarrow \mathbf{\Psi}_{G_1}$$

## 4 Proof of Theorem 4

**Lemma 1 (Eigenvalue Interlacing Lemma [3])** *Let $A^{'} = A + \sigma zz^T$ be the rank one perturbation of $A$ matrix with $||z||_2 = 1$. Then, the eigenvalues of $A^{'}$ will interlace with eigenvalues of $A$ such that if $\sigma \geq 0$, $\lambda_0 \leq \lambda^{'}_0 \leq \lambda_1 \leq \lambda^{'}_1 \leq .... \leq \lambda_{N-1} \leq \lambda^{'}_{N-1} \leq \lambda_{N-1} + \sigma$; or otherwise (i.e., if $\sigma < 0$), $\lambda_0 + \sigma \leq \lambda^{'}_0 \leq \lambda_0 \leq \lambda^{'}_1 \leq \lambda_1 \leq .... \leq \lambda^{'}_{N-1} \leq \lambda_{N-1}$.*

Let $\mathcal{S}_{xy}$ and $\mathcal{S}^{'}_{xy}$ be the graph $f$-spectral distance before and after the single edge perturbation or rank one modification of graph Laplacian $L$. Then, the modified graph Laplacian $L^{'}$ can be expressed as, $L^{'} = L + 2\triangle w_{ij}ee^T$, where $w^{'}_{ij} = w_{ij} + \triangle w_{ij}$ is the modification of weight edge $w_{ij}(\geq 0)$ to $w^{'}_{ij}(\geq 0)$ and $e$ is $N$ size column vector with $e_i = \frac{1}{\sqrt{2}}, e_j = -\frac{1}{\sqrt{2}}$ and $0$ otherwise. Since our goal is to inspect the dependency from $f(\lambda)$ perspective, therefore, we will eliminate any dependency on the Laplacian eigenvectors by bounding them. We will focus our analysis on $\triangle w_{ij} \geq 0$ and note that here $\sigma = 2\triangle w_{ij}$.

**Case I**: $f$ is an increasing function and $\mathcal{S}^{'}_{xy} > \mathcal{S}_{xy}$,

$$|\mathcal{S}^{'}_{xy} - \mathcal{S}_{xy}| = \Big| \sum_{k=0}^{N-1} f(\lambda^{'}_k)(\phi^{'}_k(x) - \phi^{'}_k(y))^2 - \sum_{k=0}^{N-1} f(\lambda_k)(\phi_k(x) - \phi_k(y))^2 \Big|$$

Applying Lemma 1, and using the fact that rows of eigenvectors matrix are orthogonal to each other and have unit norm, we get,

$$\triangle \mathcal{S} \leq \left| f(\lambda_{N-1} + \sigma) \sum_{k=0}^{N-1} (\phi_k'(x) - \phi_k'(y))^2 - f(\lambda_1) \sum_{k=0}^{N-1} (\phi_k(x) - \phi_k(y))^2 \right|$$

$$= 2 \left| f(\lambda_{N-1} + \sigma) - f(\lambda_1) \right|$$

**Case II**: $f$ is an increasing function and $\mathcal{S}_{xy}' < \mathcal{S}_{xy}$. Again applying Lemma 1, and using orthonormal property of eigenvector, matrix we get,

$$|\mathcal{S}_{xy} - \mathcal{S}_{xy}'| = \left| \sum_{i=0}^{N-1} f(\lambda_k)(\phi_i(x) - \phi_i(y))^2 - \sum_{i=0}^{N-1} f(\lambda_k')(\phi_i'(x) - \phi_i'(y))^2 \right|$$

$$\leq \left| f(\lambda_{N-1}) \sum_{i=0}^{N-1} (\phi_i(x) - \phi_i(y))^2 - f(\lambda_1') \sum_{i=0}^{N-1} (\phi_i'(x) - \phi_i'(y))^2 \right|$$

$$= 2 \left| f(\lambda_{N-1}) - f(\lambda_1) \right|$$

Combining **Case I** and **Case II** together, we get following bound for $f$ as increasing function,

$$\triangle \mathcal{S} \leq 2|f(\lambda_{N-1} + 2\triangle w_{ij}) - f(\lambda_1)|$$

**Case III**: $f$ is a decreasing function of $\lambda$. Same results hold and can be derived in similar fashion.

$$\triangle \mathcal{S} \leq 2|f(\lambda_{N-1} + 2\triangle w_{ij}) - f(\lambda_1)|$$

Note: The above results are also valid in the case of normalized graph Laplacian $L_{norm}$. Lastly, when the weight change $\Delta w$ is negative (and constrained within a certain range), the stability bounds are similar but loose when compared to the positive $\Delta w$ change case. The bounds are loose due to the fact that the Eigenvalue Interlacing Lemma does not explicitly provide the bound on the change in the largest eigenvalue with respect to negative $\Delta w$ change. As a result, the change in $f$-spectral distance (i.e., $\Delta \mathcal{S}_{xy}$) is loosely upper bounded by $2|f(\lambda_{N-1})|$ in case of an increasing function while for a decreasing function, it is upper bounded by $2|f(\lambda_1)|$.

## 5   Proof of Theorem 5

**Lemma 2 (McDiarmid's Inequality [7])** *Let* $\mathbf{X} = (x_1, x_2, ..., x_m)$ *be a set random variables and* $F : \mathbf{X} \to R$ *and* $x_i'$ *be the substitution of* $x_i$. *Now if,*

$$\sup_{x_1,..,x_i,..,x_m,x_i'} |F(x_1,..,x_i,..,x_m) - F(x_1,..,x_i',..,x_m)| \leq c_i$$

$$= \sup_{x_1,..,x_i,..,x_m,x_i'} |F_X - F_{X^i}| \leq c_i \quad, \forall i$$

*Then, the following exponential bound holds,*

$$P\left(F(X) - \mathbf{E}_X[F(X)] \geq \epsilon\right) \leq e^{-\frac{2\epsilon^2}{\sum_{i=1}^m c_i^2}}$$

Consider the weighted adjacency matrix entries as random variables, and let $\mathcal{D}$ be the distribution from which weights are independently sampled. We fix the order of $\frac{n(n-1)}{2}$ random variables in weighted adjacency matrix as $\left\{w_1, w_2, ..., w_{\frac{N(N-1)}{2}}\right\}$ and assume weights are bounded $0 < \alpha \leq w_i \leq \beta, \forall i$. Our goal is to to obtain exponential bounds on the expected value of $\mathcal{S}(x, y)$. Now, we have $\mathcal{S}(x, y)$ as the function of $f(L)$ i.e., $\mathcal{S}_f(x, y) = F(f(L)) = F(f(D - W)) = F(w_1, w_2, ..., w_{\frac{N(N-1)}{2}})$. Then, in order to apply McDiarmid's inequality, we bound the following,

| Laplacian | $f$ as increasing function | $f$ as decreasing function |
|-----------|---------------------------|----------------------------|
| $L$ | $2f(2\beta N + 2\beta)$ | $2f(\alpha N)$ |
| $L_{norm}$ | $2f(2 + 2\beta)$ | $2f(\alpha N)$ |

Table 1: Bounds on $\Theta$.

$$
\sup|F - F^i| = \sup_{w_1,..,w_{\frac{N(N-1)}{2}},w_i'} \left| F\left(w_1,..,w_i,..,w_{\frac{N(N-1)}{2}}\right) - F\left(w_1,..,w_i',..,w_{\frac{N(N-1)}{2}}\right) \right|
$$

$$
\leq \sup_{\lambda_{N-1},\lambda_1,\triangle w_i} \triangle\mathcal{S} \qquad \text{(Using Theorem 4)}
$$

$$
= \Theta
$$

Then, applying applying Lemma 2 gives the following bound with probability $1 - \delta$, where $\delta \in (0, 1)$.

$$
P\left(\mathcal{S}_f(x,y) - \mathbf{E}[\mathcal{S}_f(x,y)] \geq \epsilon\right) \leq e^{-\frac{4\epsilon^2}{N(N-1)\Theta^2}}
$$

$$
\implies \left|\mathcal{S}_f(x,y) - \mathbf{E}[\mathcal{S}_f(x,y)]\right| \leq \frac{\Theta}{2}\sqrt{N(N-1)}\sqrt{\log\frac{1}{\delta}}
$$

Same results also hold for $L_{norm}$ case.

Now, $\Theta$ itself depends upon the bounds on $\lambda_1$, $\lambda_{N-1}$ and $\triangle w_i$. Let $\lambda_{N-1}^{up}$ is the largest upper bound on $\lambda_{N-1}$ and $\lambda_1^{low}$ is the smallest lower bound possible over all graphs of size $N$. Since $w_i > 0\ \forall i$, we have a complete graph. Now according to Lemma 1, $\lambda_1$ eigenvalue will always increase, with increase in edge weights. As a result, $\lambda_1^{low}$ will achieve it lowest value when all weights are equal to $\alpha$ which implies $\lambda_1^{low} \geq \alpha N$.

**Case I**: Consider Laplacian as $L$ and $f$ as increasing function. Then, $\lambda_{N-1}^{up} \leq 2\beta N$ [1].

$$
\triangle\mathcal{S} \leq 2|f(\lambda_{N-1} + 2\triangle w_i) - f(\lambda_1)|
$$

$$
\leq 2|f(\lambda_{N-1}^{up} + 2\triangle w_i)|
$$

$$
= 2|f(2\beta N + 2\beta)|
$$

For decreasing $f$ function, we have,

$$
\triangle\mathcal{S} \leq 2|f(\lambda_1) - f(\lambda_{N-1} + 2\triangle w_i)|
$$

$$
\leq 2|f(\lambda_{N-1}^{low})|
$$

$$
= 2|f(\alpha N)|
$$

**Case II**: Consider Laplacian as $L_{norm}$ and $f$ as increasing function. Then, $\lambda_{N-1}^{up} \leq 2$ and,

$$
\triangle\mathcal{S} \leq 2|f(2 + 2\beta)|
$$

For decreasing $f$ function, we have,

$$
\triangle\mathcal{S} \leq 2|f(\alpha N)|
$$

The summary of $\Theta$ bounds are given in Table 1 for different cases. *This completes the proof.*

**Connection to Uniform Stability**: The $f$-spectral distance function can be thought of as a learning algorithm $F$ on a graph which satisfies the notion of uniform stability given as 2, $\sup_{X,\mathbf{z}}\big|\ell(A_X, \mathbf{z}) -$

$\ell(A_{X^i}, \mathbf{z})\big| \leq \eta$, where $\ell$ is the loss function, $A$ is the learning algorithm, $\mathbf{z}$ is any data point and $X$ is the set of $N$ data points. Uniform stability is a strong stability criteria which provides generalization guarantees by establishing an upper bound $\eta$ on the change in loss due to the modification of a single data point from the training set (and by subsequently taking the supremum over all possible training sets $X$ and $z$ data point). To conjure the notion of uniform stability for distance function operating on a graph, we can replace $A$ by $F$ (distance function), $\ell$ by identity function, $X$ by $|E|$ set of edges in the graph and define $X^i$ as the modification of any single edge and replace $\mathbf{z}$ as the pairwise node input $(x, y)$. Then in case of $f$-spectral distance function, $\eta$ is nothing but equal to $\Theta$ and thus, satisfies notion of uniform stability in the sense defined above.

## 6  Experiments and Results

| Datasets | MUTAG | PROTEIN | D&D |
|---|---|---|---|
| Harmonic | **92.12** | **73.42** | **77.10** |
| Biharmonic | 89.24 | 70.06 | 75.34 |

Table 2: Preliminary Experiment: Classification accuracy on few bioinformatics datasets. Harmonic based feature space yields higher accuracy than biharmonic due to sparseness.

**Parameters Selection**: For Random-Walk (RW) kernel, decay factor is chosen from $\{10^{-6}, 10^{-5}..., 10^{-1}\}$. For Weisfeiler-Lehman (WL) kernel, we chose $h = 2$ as the maximum iteration to limit the exponentially time increase and feature space of the kernel and in case of unlabeled classification, we fed node degree as the node labeled data. For graphlet kernel (GK), we chose graphlets size $\{3, 5, 7\}$. For deep graph kernels (DGK), the window size and dimension is taken from the set $\{2, 5, 10, 25, 50\}$ and report the best classification accuracy obtained among: deep graphlet kernel, deep shortest path kernel and deep Weisfeiler-Lehman kernel. For Multiscale Laplacian Graph (MLG) kernel, we chose $\eta$ and $\gamma$ parameter of the algorithm from $\{0.01, 0.1, 1\}$, radius size from $\{1, 2, 3, 4\}$, and level number from $\{1, 2, 3, 4\}$. While in case of unlabeled classification, we provide degree of the node as the labeled data. For diffusion-convolutional neural networks (DCNN), we chose number of hops from $\{2, 5\}$ and AdaGrad algorithm (gradient descent) with parameters: learning rate 0.05, batch size 100 and number of epochs 500. Again, node degree served as labeled data for the case of unlabeled classification. For the rest, best reported results were borrowed from PATCHY-SAN (CNNs) [8], skewed graph spectrum (SGS) [5] and graphlet spectrum (GS) [6] papers, since the experimental setup was the same and a fair comparison can be made.