[Reviews · NeurIPS 2017]

Reviewer 1



The authors propose a kernel for unlabeled graphs based on the histogram of the pairwise node distances where the notion of distance is defined via the Eigen decomposition of the laplacian matrix. They show that such a notion enjoys useful uniqueness and stability properties, and that when viewed as an embedding it is a generalization of known graph embedding and dimensional reduction approaches. The work would benefit from a clearer exposition (briefly anticipating the notion of "distance" or of invariance, uniqueness, stability and sparsity albeit in an informal way). Figure 3 needs to be rethought as it is not informative in the current form (perhaps showing the histogram of the number of non zero elements in the two cases?) The authors should be careful not to oversell the results of Theorem 1, as in practice it is not known how does the number of non unique harmonic spectra increases with the increase in graph size (after all graphs in real world cases have sizes well exceeding 10 nodes). Also it is not clear whether having near identical spectra could be compatible with large structural changes as measured by other sensible graph metrics. Theorem 5 in the current form is not informative as it encapsulates in \theta a variability that is hard for the reader to gauge. The stability analysis seems to be based on the condition that the variation of weights be positive; perhaps the authors want to add a brief comment on what could be expected if the variation is unrestricted? An empirical analysis of the effect of the histogram bin size should probably be included. The work of Costa and De Grave. "Fast neighborhood subgraph pairwise distance kernel." ICML 2010, seems to be related as they use the histogram of (shortest path) distances between (subgraphs centered around) each pair of nodes, and should probably be referenced and/or compared with. There are a few typos and badly formatted references. However the work is of interest and of potential impact.

Reviewer 2



SUMMARY This paper studies the properties of features extracted from graph spectral decomposition. The procedure to compute the proposed method: FGSD (family of graph spectral distances) is as follows. Given some function f and a graph, distance between nodes (x and y) in a graph is given as S_f as in equation (1). Then the graph representation R_f of g is obtained as the histogram of distribution of edges in feature space. S_f (or FGSD) essentially marginalizes distance between x and y in graph spectrum with a weight function f. Depending on f, S_f is shown to be able encode either global or local structure of a given graph. FGSD is also shown to possess uniqueness (up to permutation of vertex labels), stability with respect to eigendecomposition. Sparsity in resulting graph feature matrix is presented for f(x)=1/x, and an efficient computation of S_f using a fixed point iteration is presented. Finally, FGSD is demonstrated to show superior performance to existing graph kernels in classifying graphs. COMMENTS This paper studies graph similarity based on a weighted reconstruction of spectral decomposition. Interesting properties are derived, however, more experiments would show the detailed properties of FGSD, for example, by using synthesized graphs. In computational experiments classifying labeled graphs, FGSD is claimed to outperform graph kernels that utilize node labels. This point should be investigated more into details. For what type of graphs do FGSD perform well ? Is the node label information not essential for those tasks?

Reviewer 3



This paper proposes a family of graph spectral distances and characterizes properties of uniqueness, stability, and sparsity. The paper proposes fast approximate algorithms for computing these distances and identifies harmonic and biharmonic distances as a suitable family. Empirical evaluations support the utility of these distances. These distances will form a useful tool in featurizing and classifying graphs. The only reason for not providing a stronger accept is that I am not familiar with the relevant literature in this area.